# Significance of *TERT* Genetic Alterations and Telomere Length in Hepatocellular Carcinoma

**DOI:** 10.3390/cancers13092160

**Published:** 2021-04-30

**Authors:** Jeong-Won Jang, Jin-Seoub Kim, Hye-Seon Kim, Kwon-Yong Tak, Soon-Kyu Lee, Hee-Chul Nam, Pil-Soo Sung, Chang-Min Kim, Jin-Young Park, Si-Hyun Bae, Jong-Young Choi, Seung-Kew Yoon

**Affiliations:** 1Department of Internal Medicine, Collage of Medicine, The Catholic University of Korea, Seoul 06591, Korea; eworldcupzps@catholic.ac.kr (K.-Y.T.); blackiqq@catholic.ac.kr (S.-K.L.); hcnam@catholic.ac.kr (H.-C.N.); pssung@catholic.ac.kr (P.-S.S.); baesh@catholic.ac.kr (S.-H.B.); jychoi@catholic.ac.kr (J.-Y.C.); yoonsk@catholic.ac.kr (S.-K.Y.); 2The Catholic University Liver Research Center, Department of Biomedicine & Health Sciences, Collage of Medicine, The Catholic University of Korea, Seoul 06591, Korea; topiary@catholic.ac.kr (J.-S.K.); clover@catholic.ac.kr (H.-S.K.); 3Department of Research & Business Development, CbsBioscience Inc., Deajeon 34113, Korea; kcm3879@cbsbio.com (C.-M.K.); jonnypary@cbsbio.com (J.-Y.P.)

**Keywords:** telomere, *TERT*, liver neoplasm, biomarkers, treatment outcome

## Abstract

**Simple Summary:**

Telomerase reverse transcriptase (*TERT*) mutations are the most frequent genetic alterations in hepatocellular carcinoma (HCC). However, integrative analysis studies of *TERT*-telomere signaling during hepatocarcinogenesis are lacking. In this study, we investigated the clinicopathological association and prognostic value of *TERT* gene alterations and telomere length in HCC patients undergoing hepatectomy as well as transarterial chemotherapy (TACE). We found that there are eight key *TERT*-interacting genes and higher *TERT* expression and longer telomere length in HCC. We also found *TERT*-telomeric signals related to correlation with tumor differentiation and stage progression. *TERT* promoter mutations were an independent predictor of worse overall survival after hepatectomy, while *TERT* expression independently predicted worse time to progression after TACE. Telomere length was also associated with survival in TACE-treated patients. These findings suggest that *TERT*-telomere signals might be useful biomarkers for HCC, but the prognostic values may differ with tumor characteristics and treatment.

**Abstract:**

Telomerase reverse transcriptase (*TERT*) mutations are reportedly the most frequent somatic genetic alterations in hepatocellular carcinoma (HCC). An integrative analysis of *TERT*-telomere signaling during hepatocarcinogenesis is lacking. This study aimed to investigate the clinicopathological association and prognostic value of *TERT* gene alterations and telomere length in HCC patients undergoing hepatectomy as well as transarterial chemotherapy (TACE). *TERT* promoter mutation, expression, and telomere length were analyzed by Sanger sequencing and real-time PCR in 305 tissue samples. Protein–protein interaction (PPI) analysis was performed to identify a set of genes that physically interact with *TERT*. The PPI analysis identified eight key *TERT*-interacting genes, namely *CCT5*, *TUBA1B*, *mTOR*, *RPS6KB1*, *AKT1*, *WHAZ*, *YWHAQ*, and *TERT*. Among these, *TERT* was the most strongly differentially expressed gene. *TERT* promoter mutations were more frequent, *TERT* expression was significantly higher, and telomere length was longer in tumors versus non-tumors. *TERT* promoter mutations were most frequent in HCV-related HCCs and less frequent in HBV-related HCCs. *TERT* promoter mutations were associated with higher *TERT* levels and longer telomere length and were an independent predictor of worse overall survival after hepatectomy. *TERT* expression was positively correlated with tumor differentiation and stage progression, and independently predicted shorter time to progression after TACE. The *TERT*-telomere network may have a crucial role in the development and progression of HCC. *TERT*-telomere abnormalities might serve as useful biomarkers for HCC, but the prognostic values may differ with tumor characteristics and treatment.

## 1. Introduction

Hepatocellular carcinoma (HCC) is the most common primary liver cancer and the fifth leading cause of cancer-related mortality worldwide [1]. The major risk factors for HCC include liver cirrhosis, hepatitis B or C virus infection, alcohol abuse, nonalcoholic steatohepatitis, and metabolic disease. The mutation landscape in liver carcinogenesis is reportedly complicated, involving a number of pathways as well as somatic mutations in a multitude of genes [2,3,4]. Among the genetic alterations, telomerase reverse transcriptase (*TERT*) promoter mutations were reported to occur early and most frequently, affecting approximately 30–60% of all HCC patients [5,6]. Two hotspot mutations at −124 and −146 positions from the ATG start site in the *TERT* promoter have been shown to regulate *TERT* expression or the telomerase activation of human malignancies, including HCC [6,7].

Telomeres, repetitive DNA sequences (TTAGGG in vertebrates) found at the ends of the chromosome, are gradually shortened by each cell division in somatic cells, finally reaching senescence or apoptosis [8,9]. Telomeres are elongated by telomerase, a ribonucleoprotein–reverse transcriptase complex that uses its RNA as a template for the addition of simple telomeric repeats. Telomerase activity is regulated by the telomere-binding protein complex, called shelterin, which is composed of six proteins, including the telomeric repeat-binding factors (TRF) 1 and TRF2, the TRF1-interacting protein 2 (TIN2), protection of telomeres 1 (POT1), the POT1–TIN2 organizing protein (TPP1), and repressor/activator protein 1 (RAP1) [8]. However, embryonic stem cells and most cancer cells can maintain telomeres to overcome cell senescence or apoptosis. This process is controlled by *TERT*, the catalytic component of the telomerase complex that maintains telomere ends by addition of the telomere repeat TTAGGG [4,9]. Thus, *TERT* plays an important role in oncogenesis and the immortality of cancer cells.

Abnormalities in *TERT* expression or promoter mutations in HCC have been sporadically studied and reported to be associated with cancer recurrence and progression [10,11,12,13,14]. However, the clinical implications and molecular mechanisms underlying HCC initiation and progression are still unclear because only limited studies have been conducted and the previous studies included only a small number of patients or were limited only to the surgical setting of early-stage HCC. Thus, there is insufficient knowledge about how and what changes in *TERT* occur from early- to late-stage HCC. Moreover, no studies have attempted to evaluate the impact of the *TERT*-telomere network and alterations on the outcomes of various stages of HCC treated by surgical and non-surgical options.

To address these issues, the present study correlated the genetic alterations in *TERT*, gene expression, and telomere length with the clinicopathological features of HCC. In addition, the potential roles of the *TERT*-telomere network as a prognostic biomarker within the setting of hepatectomy and transarterial chemoembolization (TACE) were evaluated and compared.

## 2. Results

### 2.1. Baseline Characteristics

A total of 305 liver tissue samples were obtained from the 205 patients, including 105 tumor tissues only and 100 paired tumor and non-tumor tissues. The median patient age was 60 years old, and 165 patients (80.5%) were male. The causes of liver disease included hepatitis B virus (*n* = 138, 67.3%), hepatitis C virus (*n* = 16; 7.8%), and other non-viral diseases (*n* = 51; 24.9%). Most patients had Child−Pugh class A hepatic function (*n* = 167; 81.5%). The mean tumor size was 6.5 ± 4.9 cm and 94 (45.9%) patients had multiple tumors. The patients in the surgical group were younger and had less advanced HCC compared to those in the TACE group. The baseline characteristics of the patients are shown in Table 1.

### 2.2. Protein–Protein Interaction with TERT Gene Sets and Gene Expression

The protein–protein interaction (PPI) analysis was performed through CBS probe PINGS^TM^ based on the STRING database to establish a set of genes interacting with *TERT*. Functional clustering of the PPI analysis identified eight genes, including *TERT*, *AKT1*, *CCT5*, *mTOR*, *RPS6KB1*, *TUBA1B*, *YWHAQ*, and *YWHAZ*, as protein complexes related to *TERT* (Figure 1A). Functional interactions between the eight proteins are summarized in Appendix A. Within the eight genes, we performed gene expression analysis in the tumor and non-tumor paired samples. *TERT* mRNA was significantly overexpressed in the tumors compared to the non-tumors (*p* < 0.001), whereas the expression levels of the other seven genes were not significantly different between the tumors and non-tumors (Figure 1B).

Besides the genes derived from the PPI analysis, we also analyzed the shelterin complex which is known to regulate telomerase activity [8]. As a result, all shelterin components (TRF1, TRF2, POT1, TIN2, and RAP1) except TPP1 were significantly overexpressed in the non-tumors compared to the tumors (*p* < 0.001; Figure 1C). Negative correlations were seen between the expression statuses of the shelterin complex proteins, *TERT*, and telomere length (Appendix A).

### 2.3. Comparison of TERT and Telomere Length in Tumors versus Non-Tumors

Overall, the *TERT* gene and its function were markedly altered in the tumors compared to the adjacent non-tumor samples. Two hotspot mutations in the *TERT* promoter region, −124 C>T (C228T) and −146 C>T (C250T), were observed in 57 (27.8%) HCC samples but in only one (1.0%) non-tumor sample (*p* < 0.001; Figure 1D). Telomere lengths were assessed in 86 evaluable samples. The absolute telomere length was 39.8 ± 2.9 kb (interquartile range: 10.79–110.41), with 55.9 ± 4.1 kb for the tumor and 8.8 ± 0.6 kb for the non-tumor samples. The *TERT* expression and telomere length were significantly higher and longer, respectively, in the tumors than in the adjacent non-tumor tissues (all *p* < 0.001; Figure 1D). These findings indicate an important role of *TERT* and telomere dysfunction in hepatocarcinogenesis.

### 2.4. Alterations in TERT and Telomere Length in Relation to Clinical and Tumor Characteristics

The 57 tumoral *TERT* promoter mutations included 54 (94.7%) with −124 C>T and 3 (5.3%) with −146 C>T mutations. These two mutations were mutually exclusive in HCC. The prevalence of *TERT* promoter mutations differed according to the etiology of liver disease. The 57 patients with the mutation consisted of 32 with HBV-related HCC (23.2%), 7 with HCV-related HCC (43.8%), and 18 with NBNC (non-HBV and non-HCV) (35.3%), showing that the mutation rates were highest in HCV-related HCCs and lowest in HBV-related HCCs (Figure 2A). In the tumors, the presence of *TERT* promoter mutation may be associated with an increasing level of *TERT* expression and telomere length (Figure 2B), which suggests that tumoral *TERT* expression and telomere length were affected by the *TERT* promoter mutation status.

When analyzed by tumor characteristics, *TERT* promoter mutations tended to be more frequent in single (37/109, 33.9%) versus multiple (20/92, 21.7%) HCC (*p* = 0.056). With tumor stage progression, the *TERT* level was significantly upregulated (*p* = 0.005), whereas telomere length significantly decreased (*p* = 0.011) (Figure 2C). Regarding pathological tumor differentiation, *TERT* expression levels were positively correlated with HCC histological grade, while *TERT* promoter mutations or telomere length were not significantly different between the grades (Figure 2D).

### 2.5. TERT Abnormalities and Outcome after Hepatectomy

Patients with *TERT* promoter mutations had significantly worse overall survival (OS) than those without (*p* = 0.024; Figure 3A). Together with tumor extent, *TERT* promoter mutations (hazard ratio (HR) = 4.24, 95% CI: 1.75–10.26, *p* = 0.001) were identified as an independent factor for OS after hepatectomy (Table 2). Regarding post-operative recurrence, patients with *TERT* promoter mutations tended to suffer from earlier cancer recurrence than those without, but the difference did not reach statistical significance (*p* = 0.159; Figure 3B). Interestingly, although not statistically significant in the univariate analysis, *TERT* promoter mutations (HR = 2.98, 95% CI: 1.01–8.33, *p* = 0.048) were finally identified as an independent factor for recurrence after hepatectomy in the multivariate analysis (Table 2). The prognostic performance of *TERT* mutations was better in patients with non-HBV-related HCC than in those with HBV-related HCC (Appendix A). Unlike *TERT* mutations, *TERT* expression levels or telomere length had no significant impacts on OS or recurrence after hepatectomy (Figure 3C–F).

### 2.6. TERT Abnormalities and Outcome after TACE

The clinical outcomes according to *TERT* factors in patients undergoing TACE are depicted in Figure 4A–F. High *TERT* levels were associated with significantly shorter time to progression (TTP) (*p* = 0.0081) in the TACE group (Figure 4D). Together with the tumor stage, high levels of *TERT* expression (HR = 2.06, 95% CI: 1.06–4.01, *p* = 0.033) were identified as an independent predictor of shorter TTP after TACE (Table 3). Unlike its predictive value after hepatectomy, the status of *TERT* promoter mutations had no impact on OS or TTP after TACE (Figure 4A,B).

## 3. Discussion

The current study, involving a large cohort of patients, evaluated the clinicopathological association of *TERT* abnormalities and their biomarker functions in predicting HCC outcomes. In our results, the clinicopathological association and patterns of the *TERT*-telomere network varied substantially with the underlying disease, tumor stage, and differentiation. The prevalence of *TERT* promoter mutations was associated with significantly poorer OS and a trend toward an increased risk of HCC recurrence after hepatic resection. In addition, higher *TERT* expression was associated with advancing tumor stage and cell differentiation and shorter TTP and OS after TACE. Telomere length was correlated with *TERT* alterations and tumor characteristics. The overall findings suggest the crucial oncogenic role of *TERT*-telomere abnormalities and their utility as a prognostic factor for the outcome of HCC patients. To the best of our knowledge, our study was the first to provide a comprehensive analysis of the *TERT*-telomere network in settings of both surgical and non-surgical treatments of HCC.

In our analysis, *TERT* was the strongest differentially expressed gene among the *TERT*-interacting genes identified by the PPI analysis. All of the *TERT*-related markers such as *TERT* expression, promoter mutations, and telomere length were markedly enhanced in the tumors versus the adjacent non-tumorous tissues. In contrast, the expression of shelterin components was significantly reduced in the tumors compared to the non-tumors, indicating that the shelterin complex functions as a negative regulator of telomerase [8]. These findings again underline the fundamental role of *TERT* and telomere biology in hepatocarcinogenesis.

An interesting finding of our study was that the predictive role of *TERT* promoter mutations appeared to differ with treatment, with apparent effects on the post-surgical outcomes but not on those of non-surgical treatment. Although *TERT* promoter mutations have been found in preneoplastic lesions or early-stage HCCs as a gatekeeper event [15,16], its frequency reportedly did not appear to further increase linearly with disease progression after the establishment of HCC. In a study examining hepatic nodules development in cirrhosis, the hotspot *TERT* promoter mutations were detected in 6% of the low-grade dysplastic nodules, 19% of the high-grade dysplastic nodules, 61% of early HCC, and 41% of established HCC [15]. Indeed, our study showed a slightly decreasing trend in the frequency of promoter mutations from early- to advanced-stage HCC and a poor correlation with the outcome of patients with later-stage HCC eligible for TACE. Thus, it is presumed that the biomarker function of *TERT* mutations might be more apparent in early-stage HCC but was gradually outweighed by the tumor-promoting effects of other concomitant driver mutations emerging during HCC progression.

The prevalence of *TERT* promoter mutations was significantly different according to the underlying liver disease, with the highest frequency in HCV-related HCC (44%) followed by non-viral (38%) and HBV-related HCC (23%). The mutation rates were largely consistent with those in previous studies, which reported mutation frequencies of 50–60% in HCV-related and 25–35% in HBV-related HCCs [9,10,11]. Interestingly, we observed the more apparent association of *TERT* mutations with the prognosis of non-HBV HCC rather than HBV-related HCCs in the surgical group (Appendix A). However, due to the insufficient number of patients in each subgroup, the interesting issues of whether the prognostic performance of *TERT* mutations may vary with tumor stage, treatment, and/or the cause of HCC should be confirmed in larger studies.

Another *TERT* marker, *TERT* gene expression, was significantly associated with worse prognosis after TACE but not after surgical resection. The observation that higher *TERT* expression levels were an independent factor predicting shorter TTP suggests the tumor-promoting effects of *TERT* reactivation [17]. Of note, there was no significant association between *TERT* promoter mutations and *TERT* expression in our data (Figure 2B). Nevertheless, *TERT* expression became significantly enhanced with the progression of tumor stage and tissue differentiation, which was eventually linked to tumor progression in TACE-treated patients. *TERT* mutations were reported to correlate with tumor initiation, whereas other mutations, such as those in *TP53* or *CTNNB1*, were associated with later stages of HCC, causing further genomic modifications [15]. Since, in our study, advanced-stage HCC harbored fewer *TERT* mutations compared to early-stage HCC, the higher *TERT* expression observed in advanced HCC might be a consequence of other telomerase-reactivating mechanisms, including HBV integration into the *TERT* sequence, *TERT* gene amplification, or the accumulation of oncogenic pathways with tumor growth, which are distinct from the mechanisms of *TERT* promoter mutations [6,17]. Taken together, the overall findings on *TERT* alterations imply that *TERT* promoter mutations would better predict the outcome of patients with early-stage HCC treated with surgery, whereas *TERT* expression may be more associated with the prognosis of later-stage HCC patients eligible for non-surgical treatments.

Together with *TERT* genetic alterations, telomere length was significantly longer in tumors versus non-tumors. Intriguingly, telomere length was shorter with tumor stage progression (Figure 2C), indicating that moderate genomic instability elicited by shortened telomeres might be advantageous to cancer evolution in advanced HCC [8,18]. Moreover, despite the absence of a correlation between relative telomere length and post-surgical outcomes, telomere length appeared potentially predictive of outcomes within the TACE-treated patients (Figure 4E,F). Our results are inconsistent with a study showing an association between telomere length and survival after hepatectomy, but consistent in that the study results suggest the potential prognostic role of telomere length in a subset of HCC patients [11]. Telomere length is under the control of the telomerase and shelterin complex. It was reported that some shelterin components were associated with HCC expressing stemness markers and that their expression was dependent on the cause of liver disease [19,20]. However, very limited information is available about the impact of shelterin complex on the outcome of HCC. The regulation of telomere length in relation to shelterin and *TERT* alterations in HCC also remains largely unknown and requires further investigation.

Our study had several limitations. It was a retrospective analysis and inevitably subject to selection bias. Thus, our findings need to be validated in large-scale prospective cohort studies. The majority (68%) of our patients had HBV-related HCC, with fewer *TERT* mutations compared to other etiologies. Thus, our data should be evaluated in different ethnic groups for further generalization. The TACE group was heterogeneous in HCC stages and not limited to intermediate HCC but also included advanced HCC patients, for which other treatments are currently indicated. The function of shelterin complex was incompletely studied regarding the protein expression and its prognostic implication in HCC. Nevertheless, the current study represented an integrative analysis of *TERT*-related factors, including *TERT* expression, promoter mutations, and telomere length across patients undergoing not only surgical resection but also non-surgical treatments and, thus, provides a more comprehensive understanding of the entire scope of *TERT*-telomere biology in hepatocarcinogenesis.

## 4. Materials and Methods

### 4.1. Patients and Treatment

A total of 205 patients who were diagnosed with HCC at the Catholic University of Korea and the Catholic Central Biobank between March 2011 to February 2019 were analyzed. HCC was diagnosed based on histological evidence, α-fetoprotein levels, or typical radiological findings according to the Korean National Cancer Center (KNCC) guidelines [21]. Histological grading of HCC was performed using the Edmonson and Steiner grading scheme [22]. Tumor stage was classified according to the modified Union for International Cancer Control (mUICC) stage endorsed by the KNCC guidelines [21]. Treatment for HCC was performed based on tumor stage and liver function according to the KNCC practice guidelines [21]. Briefly, the patients were offered surgical resection if their tumors were resectable and they had acceptable liver function. TACE using doxorubicin was offered to patients who had unresectable or multifocal HCCs. Anticancer treatments were categorized into surgical (hepatectomy) and non-surgical options (TACE-based treatments). Treatment for recurrent or refractory tumors after initial therapy was decided by multidisciplinary decision-making and the KNCC guidelines [21]. This study was approved by the Ethics Committees of The Catholic University of Korea and all other participating institutions in accordance with the 1975 Declaration of Helsinki. The patients provided informed consent to participate in the study.

### 4.2. Protein–Protein Interaction Methods

Protein–protein interactions were analyzed using CBS probe PINGS^TM^ (Protein Interaction Network Generation System, KR100957386B1; Daejon, Korea) to identify the genes interacting with *TERT*. CBS probe PINGS^TM^ uses five modules (protein–protein interactions, Path-finder, Path-linker, Path-maker, and Path-lister) to identify interacting genes, interaction distance, and interaction frequency [23].

A multi-functional analytical tool, CBS Probe PINGS^TM^, was used to match *TERT* gene with its Entrez Gene record (NCBI ID, https://www.ncbi.nlm.nih.gov/gene, accessed on 17 November 2017) from the iProClass (https://www.ncbi.nlm.nih.gov/pubmed/15022647, accessed on 17 November 2017) database, and with gene names and synonyms in UniProtKB/Swiss-Prot (Uniprot Knowledgebase, https://www.ncbi.nlm.nih.gov/pubmed/27899622, accessed on 17 November 2017) to further interchange with identification factor “Uniprot Ac” in CBS Probe PINGS^TM^. We then conducted the interactive proteins network analysis leveraging IntAct (IntAct, http://europepmc.org/abstract/MED/24234451, accessed on 17 November 2017), BioGRID (Biological General Repository for Interaction Datasets, https://www.ncbi.nlm.nih.gov/pubmed/30476227, accessed on 17 November 2017), DIP (Database of interacting proteins, https://www.ncbi.nlm.nih.gov/pubmed/10592249, accessed on 17 November 2017), HPRD (Human Protein Reference Database, https://www.ncbi.nlm.nih.gov/pubmed/18988627, accessed on 17 November 2017), and MINT (The Molecular INTeraction, https://www.ncbi.nlm.nih.gov/pmc/articles/PMC1751541, accessed on 17 November 2017) database accordingly. The selectable identification includes interaction distance, interaction type, interaction detection method, number of interactive information-related databases, number of related literature studies, and number of interaction detection methods. In this study, we investigated the direct interacting genes with the start gene as *TERT*, and the organism has been chosen as *Homo sapiens*.

### 4.3. TERT Promoter Mutation

Genomic DNA (gDNA) was extracted from fresh frozen tissue samples using a QIAamp DNA Mini Kit (Qiagen, Hidden, Germany). Direct sequencing of the tissue samples was performed for polymerase chain reaction (PCR) amplification using the following pairs of primers encompassing the mutational hotspots. For −124 bp G>A and −146 bp G>A in the *TERT* promoter, the primers were forward 5′-CAGCGCTGCCTGAAACTC-3′ and reverse 5′-GTCCTGCCCCTTCACCTT-3′. PCR was performed using a DNA Engine Tetrad 2 Peltier Thermal Cycler (Bio-Rad, Hercules, CA, USA). The sequencing data were analyzed on an ABI PRISM 3730XL Analyzer (Applied Biosystems, Foster City, CA, USA).

### 4.4. Quantitative Real-Time PCR Analysis of TERT Gene Expression and Shelterin Complex

Total RNA was extracted from 25 mg of fresh frozen tissue sample using a miRNeasy Mini Kit (Qiagen) according to the manufacturer’s instructions. *TERT* gene expression was measured by quantitative real-time PCR (qRT-PCR) using the Hs00972650_m1 TaqMan gene expression assay (Applied Biosystems, Foster City, CA, USA). The primers and probe sequences of *TERT*-interacting genes (*CCT5*, *TUBA1B*, *mTOR*, *RPS6KB1*, *AKT1*, *YWHAZ*, and *YWHAQ*) were designed in Primer Express 3.0 (Applied Biosystems). All probes were labeled with TAMRA at the 3′ end and FAM at the 5′ end (Appendix A). All measurements were normalized to the expression of the endogenous control Hs03927097_g1 GAPDH (Applied Biosystems). Relative fold-changes in *TERT* gene expression were determined by the ΔΔCT method [24]. Shelterin complex *TRF1, TRF2, POT1, TPP1, TIN2*, and *POT1* expression were also measured by qRT-PCR. The PCR conditions with the primer sequences for the six shelterin components are shown in Appendix A. All qRT-PCR assays were performed and analyzed on the ABI ViiA 7 Real-Time PCR System (Applied Biosystems).

### 4.5. Telomere Length Measurement

After gDNA extraction, telomere lengths were measured using Absolute Human Telomere Length Quantification qPCR Assay Kit (AHTLQ, Catalog #8918; ScienCell Research Laboratories, Carlsbad, CA, USA) according to the manufacturer’s instructions, as previously reported [25]. Briefly, a single copy reference primer set was used as a reference for data normalization where it recognizes and amplifies a 100 bp-long region on human chromosome 17. A reference genomic DNA sample with known telomere length was used as a reference for the calculation of telomere length of the target samples. All AHTLO assays were performed and analyzed on the CFX96 Touch™ Real-Time PCR Detection System (Bio-Rad).

### 4.6. Statistical Analysis

All data were expressed as the mean ± S.D. or median (interquartile range). Student *t*-test or Mann–Whitney U test was used to compare continuous variables, while chi-square test or Fisher’s exact test to compare categorical variables. Survival analysis was performed using the Kaplan–Meier method to estimate the cumulative rate, and the difference was evaluated based on the log-rank test. The prognostic factors for overall survival (OS) and time to progression (TTP) were performed using the Cox proportional hazard model with univariate and multivariate analysis. A side-step P-value of less than 0.05 was considered to indicate statistically significant differences. Statistical analyses were performed using IBM SPSS Statistics 20.0 (SPSS Inc., Chicago, IL, USA).

## 5. Conclusions

Our analysis of *TERT* and telomere alterations demonstrates that the *TERT*-telomere network has a crucial role in all-stage liver carcinogenesis including the development and progression of HCC, with differential *TERT* factors involved over HCC stage progression. The present findings highlight the utility of *TERT* genetic alterations and aberrant telomere biology as excellent candidate biomarkers for early diagnosis and monitoring during treatment. Future larger studies should be conducted to further evaluate whether the *TERT* pathway might serve as a potential therapeutic target and define distinct prognostic classes for HCC.

## Figures and Tables

**Figure 1 cancers-13-02160-f001:**
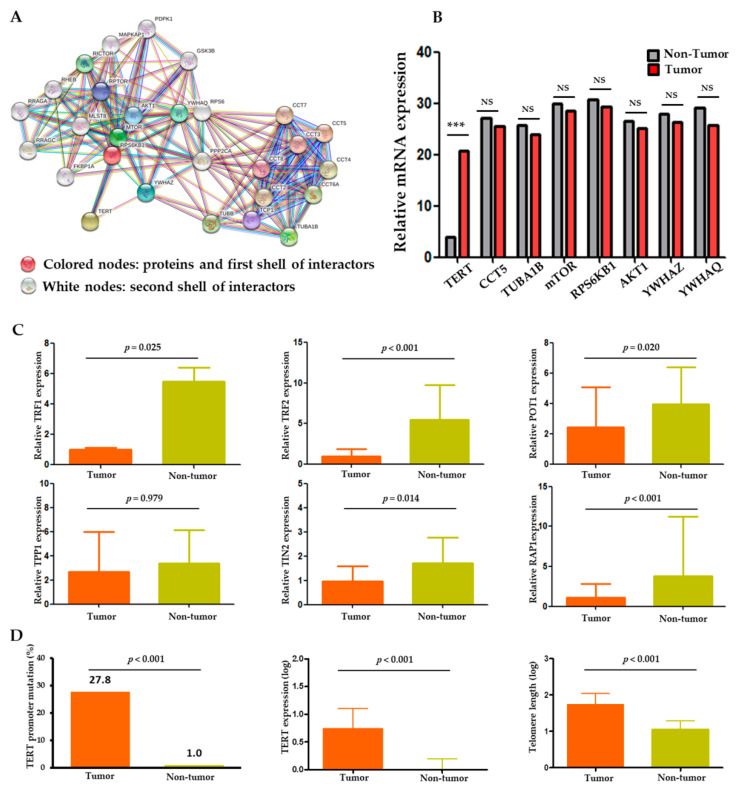
*TERT* expression and telomere length relationship in HCC. (**A**) Protein–protein interaction (PPI) analysis of *TERT* using the CBS probe PINGS^TM^ in HCC tissues. (**B**) Expression profiles of the eight *TERT*-interacting genes from the STRING database in tumor versus adjacent non-tumor tissues. (**C**) Comparison of shelterin complex TRF1, TRF2, POT1, TPP1, TINP2, and RAP1 between tumor and non-tumor tissues. (**D**) Comparison of *TERT* promoter mutations, *TERT* expression and telomere length between tumor and non-tumor tissues. TRF, telomeric repeat-binding factors; POT1, protection of telomeres 1; TPP1, POT1-TIN2 organizing protein; TIN2, TRF1 and TRF2 interacting nuclear protein 2; RAP1, repressor/activator protein 1. NS, non-significance; *** *p* < 0.001.

**Figure 2 cancers-13-02160-f002:**
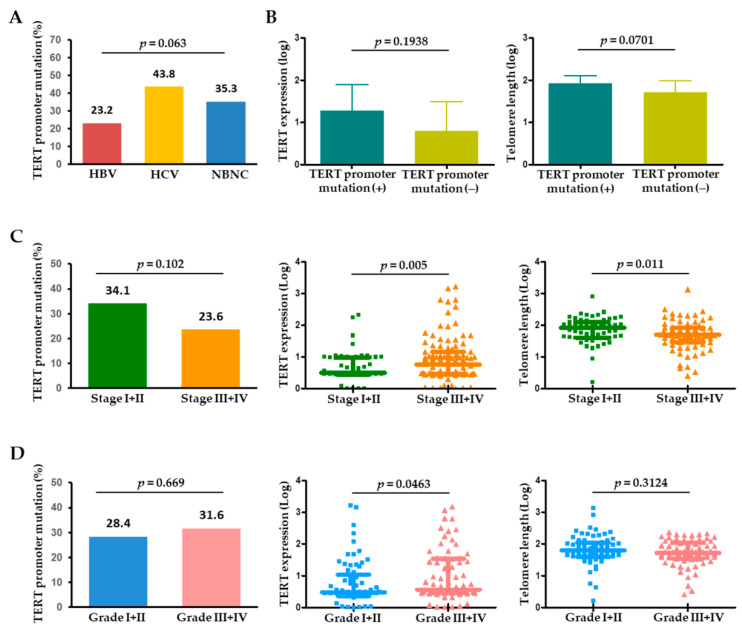
(**A**) Frequency of *TERT* promoter mutations in HCC according to the etiology of HCC. (**B**) *TERT* expression and telomere length in the presence or absence of *TERT* promoter mutations in HCC. Frequency of *TERT* promoter mutations, *TERT* expression, and telomere length according to (**C**) tumor stage and (**D**) tumor histological grade. HBV, hepatitis B virus; HCV, hepatitis C virus; NBNC, non-HBV non-HCV.

**Figure 3 cancers-13-02160-f003:**
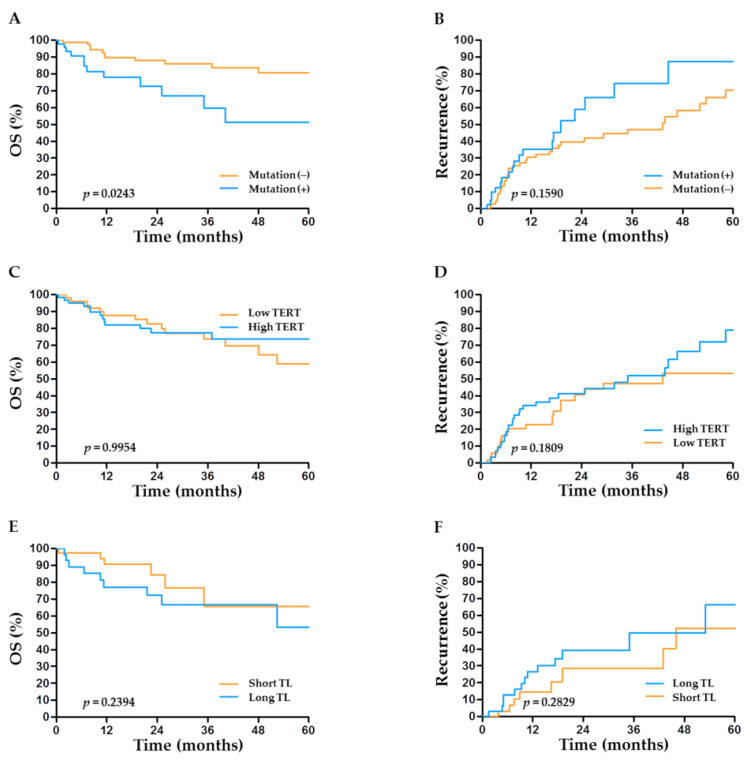
Prognostic role of *TERT* factors in the outcome of HCC after hepatectomy. Overall survival (OS) and HCC recurrence according to (**A**,**B**) the status of *TERT* promoter mutations, (**C**,**D**) the expression level of *TERT*, and (**E**,**F**) telomere length.

**Figure 4 cancers-13-02160-f004:**
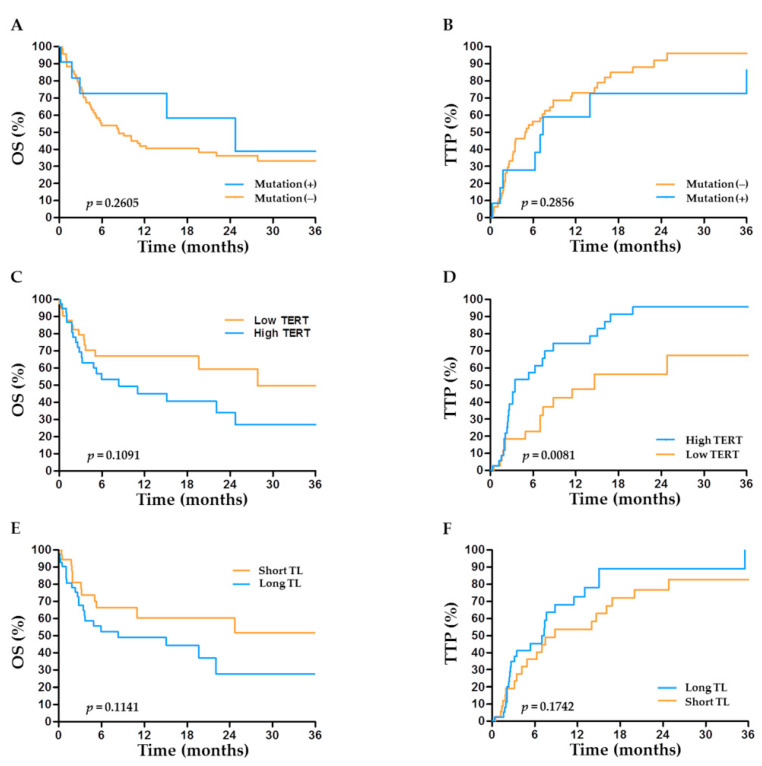
Prognostic role of *TERT* factors in the outcome of HCC after transarterial chemoembolization (TACE). Overall survival (OS) and time to progression (TTP) according to (**A**,**B**) the status of *TERT* promoter mutations, (**C**,**D**) the expression level of *TERT*, and (**E**,**F**) telomere length (TL).

**Table 1 cancers-13-02160-t001:** Baseline characteristics of the study population.

Characteristics	All Patients (*n* = 205)	Surgical Group (*n* = 121)	TACE Group (*n* = 84)
Sex			
Male	165 (80.5)	89 (73.5)	76 (90.5)
Female	40 (19.5)	32 (26.4)	8 (9.5)
Age (years)	60.0 ± 11.7	58.1 ± 11.6	64.5 ± 11.9
Cause of liver disease			
HBV	138 (67.3)	85 (70.2)	53 (63.1)
HCV	16 (7.8)	13 (10.7)	3 (3.6)
Non-viral	51 (24.9)	23 (19.0)	28 (33.3)
AST (IU/L)	45 (29.5−94.5)	45 (30−94.3)	42 (29−91)
ALT (IU/L)	34 (23−68.5)	33.5 (23−69)	33 (22−65)
Child−Pugh class			
A	167 (81.5)	111 (91.7)	56 (66.7)
B/C	38 (18.5)	10 (8.3)	28 (33.3)
Tumor size (cm)	6.5 ± 4.9	4.9 ± 4.5	8.8 ± 4.9
Tumor number			
Single	111 (54.1)	88 (72.7)	23 (27.4)
Multifocal	94 (45.9)	33 (27.3)	61 (72.6)
α-fetoprotein (ng/mL)	50.3 (5.5−800.8)	48.4 (5.4−699.1)	57.4 (5.6−881.4)
mUICC stage			
I	13 (6.3)	9 (7.4)	4 (4.8)
II	80 (39.0)	66 (54.5)	14 (16.7)
III	49 (23.9)	29 (24.0)	20 (23.8)
IV	63 (30.7)	17 (14.0)	46 (54.8)

HBV, hepatitis B virus; HCV, hepatitis C virus; AST, aspartate aminotransferase; ALT, alanine aminotransferase; mUICC, modified Union for International Cancer Control. Data are expressed as mean ± SD or median (interquartile range). Figures in parentheses indicate percentage.

**Table 2 cancers-13-02160-t002:** Prognostic variables in patients undergoing hepatic resection.

Variables	Overall Survival			Recurrence		
	Univariate	Multivariate		Univariate	Multivariate	
	*p*	HR (95% CI)	*p*	*p*	HR (95% CI)	*p*
Male sex	0.423			0.694		
Age > 60 years	0.319			0.047	5.07 (1.67−15.15)	0.004
Cause of liver disease	0.133			0.144		
AST > 90 U/L	0.353			0.473		
ALT > 60 U/L	0.357			0.595		
Child−Pugh class B/C	0.095	1.02 (0.94−1.09)	0.616	0.120		
Tumor size > 5 cm	<0.001	7.97 (3.13−20.24)	<0.001	<0.001	1.81 (0.63−5.18)	0.263
Tumor multiplicity	0.075	1.56 (0.43−5.65)	0.491	0.019	1.85 (0.44−7.71)	0.398
α-fetoprotein	0.317			<0.001	1.54 (0.56−4.23)	0.396
Tumor stage (mUICC)	<0.001	2.46 (1.45−4.16)	0.001	<0.001	2.71 (1.48−4.96)	0.001
*TERT* promoter mutation	0.029	4.24 (1.75−10.26)	0.001	0.162	2.98 (1.01−8.33)	0.048
*TERT* expression	0.913			0.362		
Telomere length	0.446			0.208		

HR, hazard ratio; CI, confidence interval; HBV, hepatitis B virus; HCV, hepatitis C virus; AST, aspartate aminotransferase; ALT, alanine aminotransferase; mUICC, modified Union for International Cancer Control; *TERT*, telomerase reverse transcriptase.

**Table 3 cancers-13-02160-t003:** Prognostic variables in patients undergoing TACE-based treatment.

Variables	Overall Survival			Time to Progression		
	Univariate	Multivariate		Univariate	Multivariate	
	*p*	HR (95% CI)	*p*	*p*	HR (95% CI)	*p*
Male sex	0.420			0.958		
Age > 60 years	0.575			0.800		
Cause of liver disease	0.150			0.291		
AST > 88 U/L	0.042	1.55 (0.49−3.12)	0.644	0.067	1.44 (0.72−2.90)	0.300
ALT > 57 U/L	0.119			0.283		
Child−Pugh class B/C	<0.001	4.55 (1.96−10.55)	0.001	0.087	1.27 (0.54−2.96)	0.576
Tumor size > 5 cm	0.001	2.68 (0.83−8.67)	0.099	0.003	2.11 (0.96−4.62)	0.061
Tumor multiplicity	0.469			0.102		
α-fetoprotein	0.003	2.67 (1.01−7.05)	0.047	0.061	1.07 (0.51−2.23)	0.855
Tumor stage (mUICC)	<0.001	1.98 (1.02−3.84)	0.043	<0.001	1.70 (1.11−2.60)	0.013
*TERT* promoter mutation	0.181			0.330		
*TERT* expression	0.028	1.58 (0.67−3.70)	0.289	0.010	2.06 (1.06−4.01)	0.033
Telomere length	0.137			0.277		

TACE, transarterial chemoembolization; HR, hazard ratio; CI, confidence interval; HBV, hepatitis B virus; HCV, hepatitis C virus; AST, aspartate aminotransferase; ALT, alanine aminotransferase; mUICC, modified Union for International Cancer Control; *TERT*, telomerase reverse transcriptase.

## Data Availability

The data presented in this study are available on request from the corresponding author. Associated clinical data cannot be provided to maintain patient confidentiality.

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
