# Peer review of "Significance of TERT Genetic Alterations and Telomere Length in Hepatocellular Carcinoma"

_cancers, 2021, doi:10.3390/cancers13092160_

Round 1
Reviewer 1 Report
All issues have now been addressed.
This manuscript is a resubmission of an earlier submission. The following is a list of the peer review reports and author responses from that submission.
Round 1
Reviewer 1 Report
The authors provide an analysis of frequency of TERT promoter mutations in their cohort of hepatocellular carcinoma, and assess for any relationship to TERT expression levels and telomere length, both in relationship to matched normal tissue, and between tumor samples with and without TERT promoter mutations. While there are marked increases in all compared to normal tissue, this finding is muted when comparing TERT mutated to TERT wildtype tumors, suggesting alternative mechanisms of increasing TERT expression and alternative mechanisms of lengthening telomeres, although these mechanisms are not explored in this manuscript. They propose that TERT promoter status and TERT expression may be predictive of value in very specific situations, with one being predictive of outcomes in patients undergoing resection but not TACE, while the other may be predictive in patients undergoing TACE but not surgical resection.
Comment #1: The methodology of the PPI interactions is unclear. Additional methodology to describe how this was performed would be useful for understanding exactly what the authors did to generate the plot.
Comment #2: Regardless, the fact that only TERT showed a significant difference between tumor and non-tumor samples calls into question the validity and translatability of the findings from the STRING database. Although the author’s analysis of the STRING database yielded eight gene sets related to TERT, only TERT showed a meaningful difference in a comparison of tumor and non-tumor samples. Is there an alternative way to validate or support clinical relevance for the findings of the analysis of the STRING database?
Comment #3: Others have looked at TERT promoter mutation in similar populations (Lee et al. Medicine 96(5):e5766), and did not find a correlation between the presence of TERT promoter mutation and survival. Can you hypothesize as to why some groups find a difference in survival while others do not?
Comment #4: For Figure 3, given the multiple comparisons made (TERT promoter mutation status, TERT expression, and telomere length, subdivided into patients by surgical treatment), there is concern that some of the findings present (most of which barely reach statistical significance) could be due to chance. It is difficult to arrive at biological reasons for why TERT promoter mutation would be predictive of survival after surgery but not TACE, while the inverse would be true for TERT expression levels. It would be helpful to see all the survival curves for all the permutations, rather than just the most promising ones, as this would allow for a better visualization of whether all the features are at least trending in the same direction. The findings would be greatly strengthened by the use of either a larger cohort to confirm the findings, or the addition of validation of these findings in an external cohort.
Minor comments: In Figure 3, what is the difference between recurrence in panel B, and PFS which is used for the analysis in D and F? Does the different terminology imply a different outcome measured? Figures S1 part B appears to have a typo on the Y-axis, and should probably be recurrence, not overall survival.
Author Response
Comments and Suggestions for Authors
The authors provide an analysis of frequency of TERT promoter mutations in their cohort of hepatocellular carcinoma, and assess for any relationship to TERT expression levels and telomere length, both in relationship to matched normal tissue, and between tumor samples with and without TERT promoter mutations. While there are marked increases in all compared to normal tissue, this finding is muted when comparing TERT mutated to TERT wildtype tumors, suggesting alternative mechanisms of increasing TERT expression and alternative mechanisms of lengthening telomeres, although these mechanisms are not explored in this manuscript. They propose that TERT promoter status and TERT expression may be predictive of value in very specific situations, with one being predictive of outcomes in patients undergoing resection but not TACE, while the other may be predictive in patients undergoing TACE but not surgical resection.
Comment #1: The methodology of the PPI interactions is unclear. Additional methodology to describe how this was performed would be useful for understanding exactly what the authors did to generate the plot.
Ans) We thank the reviewer for the helpful comment. In our research, PPI network was analyzed using CBS Probe PINGSTM with multiple databases such as iProClass, UniprotKB/Swiss-Prot, IntAct, BioGRID, DIP, HPRD, and MINT database. The detailed description of our methodology is provided in our prior paper [Park IJ, et al. Cancers 2020 Mar 26;12(4):800]. As per the reviewer’s request, we have described additional methodology of the PPI network analysis along with the citation of the study (reference #22) in the appropriate section of Method (pages 10).
Comment #2: Regardless, the fact that only TERT showed a significant difference between tumor and non-tumor samples calls into question the validity and translatability of the findings from the STRING database. Although the author’s analysis of the STRING database yielded eight gene sets related to TERT, only TERT showed a meaningful difference in a comparison of tumor and non-tumor samples. Is there an alternative way to validate or support clinical relevance for the findings of the analysis of the STRING database?
Ans) To support clinical relevance for the findings from the PINGSTM database, we would like to provide the results through the analysis of TERT and TERT gene set using Kaplan-Meier plotter (http://kmplot.com/analysis/). (The Kaplan Meier plotter is capable to assess the effect of 54k genes (mRNA, miRNA, and protein) on survival in 21 cancer types including HCC. Sources for the databases include GEO, EGA, and TCGA. Primary purpose of the tool is a meta-analysis based discovery and validation of survival biomarkers. When evaluated regarding the prognostic value of the eight gene sets using the tool, high expression of TERT, CCT5, TUBA1B, YWHAZ, and YWHAQ was significantly associated with poorer overall survival (TERT, HR=1.51, 95% CI: 1.03-2.2, P=0.032; CCT5, HR=2.77, 95% CI: 1.89-4.06, P=5.4e-08; TUBA1B, HR=2.1, 95% CI: 1.47-3.01, P=3.1e-05; YWHAZ, HR=1.8, 95% CI: 1.25-2.59, P=0.0013; YWHAQ, HR=1.98, 95% CI: 1.38-2.83, P=0.00015) in patients with HCC. However, the expression of mTOR (HR=0.68, 95% CI: 0.45-1.05, P=0.081), Akt1 (HR=0.72, 95% CI: 0.49-1.06, P=0.091) or RPS5KB1 (HR=1.32, 95% CI: 0.91-1.91, P=0.14) was not associated with the patient survival. The survival curves of HCC patients using Kaplan-Meier plotter are as follows.
In line with the comment on the aforementioned question, we have provided the detailed description of our method for PPI analysis using CBS probe PINGSTM in the method section. As the reviewer stated, in our qRT-PCR results (Fig 1B) only the TERT mRNA expression showed a meaningful difference in a comparison of tumor and non-tumor samples, suggesting that TERT deserves further investigation in patients with HCC. Therefore, our further analyses were performed, focusing primarily on the clinical relevance of TERT among the eight gene sets.
In addition, we would also provide the information on the previously published papers using our database CBS Probe PINGSTM for protein-protein interaction in the following table.
Table: Summary of the published studies using the PINGSTM database
|
SEQ |
Gene |
Uniprot ac |
Uniprot ID |
Protein Name |
Datebase |
PubMed |
URL |
Detail URL |
|
0 |
TERT |
O14746 |
TERT_HUMAN |
Telomerase reverse transcriptase |
  |
  |
  |
  |
|
1 |
AKT1 |
P31749 |
AKT1_HUMAN |
RAC-alpha serine/threonine-protein kinase |
MINT |
Pubmed:18775701 |
https://mint.bio.uniroma2.it/index.php/results-interactions/?id=MINT-6742762 |
https://mint.bio.uniroma2.it/index.php/detailed-curation/?id=MINT-6742762 |
|
  |
  |
  |
  |
  |
IntAct |
Pubmed:18775701 |
https://www.ebi.ac.uk/intact/pages/interactions/interactions.xhtml?query=MINT-6742762&searchClass=Experiment&page=1 |
https://www.ebi.ac.uk/intact/interaction/EBI-8067912 |
|
  |
  |
  |
  |
  |
BioGRID |
Pubmed:15843522,18775701,28205554 |
https://thebiogrid.org/106710/summary/homo-sapiens/akt1.html |
https://thebiogrid.org/interaction/282427/akt1-tert.html |
|
2 |
CCT5 |
P48643 |
TCPE_HUMAN |
T-complex protein 1 subunit epsilon |
BioGRID |
Pubmed:23741361 |
https://thebiogrid.org/116603/summary/homo-sapiens/cct5.html |
https://thebiogrid.org/interaction/872854 |
|
3 |
MTOR |
P42345 |
MTOR_HUMAN |
Serine/threonine-protein kinase mTOR |
BioGRID |
Pubmed:15843522 |
https://thebiogrid.org/108757/summary/homo-sapiens/mtor.html |
https://thebiogrid.org/interaction/282428 |
|
4 |
RPS6KB1 |
P23443 |
KS6B1_HUMAN |
Ribosomal protein S6 kinase beta-1 |
BioGRID |
Pubmed:15843522 |
https://thebiogrid.org/112112/summary/homo-sapiens/rps6kb1.html |
https://thebiogrid.org/interaction/282430 |
|
5 |
TUBA1B |
P68363 |
TBA1B_HUMAN |
Tubulin alpha-1B chain |
BioGRID |
Pubmed:23741361 |
https://thebiogrid.org/115651/summary/homo-sapiens/tuba1b.html |
https://thebiogrid.org/interaction/872857 |
|
6 |
YWHAQ |
P27348 |
1433T_HUMAN |
14-3-3 protein theta |
BioGRID |
Pubmed:10835362 |
https://thebiogrid.org/116168/summary/homo-sapiens/ywhaq.html |
https://thebiogrid.org/interaction/306078/tert-ywhaq.html |
|
7 |
YWHAZ |
P63104 |
1433Z_HUMAN |
14-3-3 protein zeta/delta |
BioGRID |
Pubmed:10835362 |
https://thebiogrid.org/113366/summary/homo-sapiens/ywhaz.html |
https://thebiogrid.org/interaction/306076 |
Collectively, we believe that the Kaplan-Meier plotter data could support clinical relevance for the findings from our analysis using the PINGSTM database. And we also expect that the list of published studies could enhance the validity and reliability of our assessment using CBS probe PINGSTM for PPI network analysis.
Comment #3: Others have looked at TERT promoter mutation in similar populations (Lee et al. Medicine 96(5):e5766), and did not find a correlation between the presence of TERT promoter mutation and survival. Can you hypothesize as to why some groups find a difference in survival while others do not?
Ans) We would state that among the prior studies evaluating TERT promoter mutation in HCC, almost all studies except Lee et al (Medicine 96(5):e5766) showed a significant correlation between TERT promoter mutation and survival [Kawai-Kitahata et al. J Gastroenterol 2016;51:473.; Huang et al. Oncotarget 2017;8:26288; Li et al. Theranostics 2018;8:1740.; Ako et al. Oncology 2020; 98;311.], which is consistent with the findings of our study. The reason why the study by Lee et al. has discrepant results from our and other studies might be related to patient heterogeneity in underlying liver disease, tumor biology and treatment. Although Lee et al. and our study were conducted in the same country, Lee et al. included a larger number of HCC patients with non-viral cause, accounting for 60.6% (97/160) of entire population. In contrast, our study had a minority of HCC patients with non-viral cause, who consisted of only 24.9% for the entire cohort and 19.0% for the surgical cohort. Given that most of the Korean studies represent a high proportion of virally-associated HCC (60-70% for HBV-related HCC and 10-15% for HCV-related HCC versus 25-35% for the remaining non-viral causes), the study by Lee et al. likely represents a surprisingly high proportion of non-viral cause of HCC. As evidenced in our results, the prognostic role of TERT alterations may differ with underlying liver disease. Thus, the discrepant findings may result from the different patient characteristics. Another point that we would like to make is tumor heterogeneity (intra- and inter-tumoral heterogeneity). Even within the same tumor, subclones with different biological nature (exhibiting distinct morphological and phenotypic profiles, including cellular morphology, gene expression, metabolism, proliferation, and metastatic potential) may exist. Depending on the biopsied, examined areas within the tissue, study results may vary due to the clonal heterogeneity within cancer tissue. Lastly, we would state that the effects of TERT alterations may differ with treatment settings, as shown in our analyses of the surgical and non-surgical cohort. Overall, it seems that the mechanisms of telomere and TERT regulation are very complicated and multiple regulatory mechanisms may exist in cancer initiation and progression. We thank the reviewer for this good question and have commented this point in the appropriate section of Discussion (pages 9-10).
Comment #4: For Figure 3, given the multiple comparisons made (TERT promoter mutation status, TERT expression, and telomere length, subdivided into patients by surgical treatment), there is concern that some of the findings present (most of which barely reach statistical significance) could be due to chance. It is difficult to arrive at biological reasons for why TERT promoter mutation would be predictive of survival after surgery but not TACE, while the inverse would be true for TERT expression levels. It would be helpful to see all the survival curves for all the permutations, rather than just the most promising ones, as this would allow for a better visualization of whether all the features are at least trending in the same direction. The findings would be greatly strengthened by the use of either a larger cohort to confirm the findings, or the addition of validation of these findings in an external cohort.
Ans) We thank the reviewer for the helpful comment. In this revised version, we have provided all the survival curves (in Revised Figures 3 and 4). In the revised version, listing all the survival curves clearly allows for a better visualization of the trends of TERT-related variables and enhance understanding of TERT-telomere biology in the outcomes of patients undergoing different treatments. As the reviewer pointed out, even if some did not reach statistical significance, those findings need to be validated and substantiated in future studies of a larger cohort. We have commented this point in the appropriate section of Discussion (page 9)
Minor comments:
- Q) In Figure 3, what is the difference between recurrence in panel B, and PFS which is used for the analysis in D and F? Does the different terminology imply a different outcome measured?
Ans) We agree with the reviewer’s concern. Before we respond to this question, we would acknowledge that there were an error and typo in the PFS in the prior manuscript. It should be TTP. We made corrections to them in this revision. Regarding this question, we would kindly refer the reviewer to a recent study by Yan et al. [J Hepatol 2019;70:570-571.], which has proposed that ‘time to recurrence’, but not recurrence-free survival (RFS), should be the endpoint used to predict the outcomes of HCC. Unlike time to recurrence, RFS is generally defined as ‘‘the time from date of curative surgery to the time of recurrence or death”. Because we did not count death when analyzing recurrence in the surgical group, we provide TTP (time to progression) instead of PFS (progression-free survival) for consistency in this revised version. Given that TTP is a well-known term and often used in palliative treatment, we believe that TTP in the TACE group would better match with recurrence in the surgical group, minimizing misinterpretation. In line with Comment #4, the revised version provides all the survival curves for both the surgical and non-surgical (TACE) groups (shown in the revised figures 3 and 4). Because TACE is not a curative option for HCC, the term ‘Recurrence or RFS’ may not be appropriate for the TACE group. Rather, TTP would be better used for the TACE group. We thus have provided TTP instead of RFS for the TACE group in the revised Figures 4.
- Q) Figures S1 part B appears to have a typo on the Y-axis, and should probably be recurrence, not overall survival.
Ans) We thank the reviewer for bringing this to our attention. As the reviewer pointed out, there was a typo on the Y-axis. It was of course an error. We have made changes to the wording, replacing with ‘HCC recurrence’ in the revised figure S2-part B.
Submission Date 01 February 2021
Date of this review 25 Feb 2021 18:40:58

Reviewer 2 Report
Jang et al. describe several experiments to compare several telomerase-related parameters, including TERT gene expression, TERT promoter mutations, telomere length, and telomerase interacting proteins with different aspects in HCC etiology. The experiment's conception is sound but is flawed by oversimplified assumptions and a lack of rigor in statistical analysis.
- There is a large body of data for proteins that physically interact with telomerase (e.g., shelterin) and transcriptional regulators of the TERT gene. It is not necessary to default to ontology to define such potential interactions. Rather the known interactors should be explored and are far more likely to produce interpretable and interesting results. The nature of these proteins should also be presented in the introduction. In the absence of this data, the “telomere network data” should be eliminated.
- A pervasive problem in the manuscript is the use of “marginally significant” vs. significant results. While the authors can briefly discuss trends, marginally significant use is non-standard and without statistical meaning. The conclusions that ARE borne out by the data presented are restricted to the following.
- Figures 1C, 1D, and 1E clearly show that tumors are associated with promoter mutations (interesting), telomere length increases, and TERT expression increases.
- Differences in HCC are significant for TERT expression and telomere length. This was verified by univariate analysis.
- Overall survival IS affected by promoter mutations.; Progression-free survival IS different with low TERT vs. high TERT, a result that is not unexpected.
3 An indication of the actual range of the absolute telomere length is needed to draw any inference regarding genetic stability.
Trivial textual issue: Need to define TACE., OS in the text in addition to the figure legend.
Author Response
Comments and Suggestions for Authors
Jang et al. describe several experiments to compare several telomerase-related parameters, including TERT gene expression, TERT promoter mutations, telomere length, and telomerase interacting proteins with different aspects in HCC etiology. The experiment's conception is sound but is flawed by oversimplified assumptions and a lack of rigor in statistical analysis.
Q1) There is a large body of data for proteins that physically interact with telomerase (e.g., shelterin) and transcriptional regulators of the TERT gene. It is not necessary to default to ontology to define such potential interactions. Rather the known interactors should be explored and are far more likely to produce interpretable and interesting results. The nature of these proteins should also be presented in the introduction. In the absence of this data, the “telomere network data” should be eliminated.
Ans) We thank the reviewer for the helpful comment. As recommended by the reviewer, we analyzed the expression of shelterin complex in our 68 available samples (NT=22 and T=46). As a result, the expression of the examined shelterin proteins TRF1, TRF2, and POT1 was all decreased in the tumors than in the non-tumors (Figure 1C). Scatter plots showed a negative correlation between TERT and the shelterin complex (Supplementary Figure). Given that shelterin complex acts as a negative regulator of telomerase [Okamoto et al. Cells 2019;8:107. Ref #8], these findings are consistent with Figure 1D indicating an overexpression of TERT in the tumors versus non-tumors. In addition, as per the reviewers request, we have added a brief mention of the known interactors shelterin complex proteins to the Introduction section and also provided the results on the expression and correlation of TERT, telomere, and shelterin proteins TRF1, TRF2, and POT1 in the supplementary data in the revised version.
On the other side, we agree with the reviewer’s concern on our DB-based analysis of protein-protein interaction. We would provide the list of published studies using our database CBS Probe PINGSTM for protein-protein interaction in the following table.
Table: Summary of the published studies using the PINGSTM database
|
SEQ |
Gene |
Uniprot ac |
Uniprot ID |
Protein Name |
Database |
PubMed |
URL |
Detail URL |
|
0 |
TERT |
O14746 |
TERT_HUMAN |
Telomerase reverse transcriptase |
  |
  |
  |
  |
|
1 |
AKT1 |
P31749 |
AKT1_HUMAN |
RAC-alpha serine/threonine-protein kinase |
MINT |
Pubmed:18775701 |
https://mint.bio.uniroma2.it/index.php/results-interactions/?id=MINT-6742762 |
https://mint.bio.uniroma2.it/index.php/detailed-curation/?id=MINT-6742762 |
|
  |
  |
  |
  |
  |
IntAct |
Pubmed:18775701 |
https://www.ebi.ac.uk/intact/pages/interactions/interactions.xhtml?query=MINT-6742762&searchClass=Experiment&page=1 |
https://www.ebi.ac.uk/intact/interaction/EBI-8067912 |
|
  |
  |
  |
  |
  |
BioGRID |
Pubmed:15843522,18775701,28205554 |
https://thebiogrid.org/106710/summary/homo-sapiens/akt1.html |
https://thebiogrid.org/interaction/282427/akt1-tert.html |
|
2 |
CCT5 |
P48643 |
TCPE_HUMAN |
T-complex protein 1 subunit epsilon |
BioGRID |
Pubmed:23741361 |
https://thebiogrid.org/116603/summary/homo-sapiens/cct5.html |
https://thebiogrid.org/interaction/872854 |
|
3 |
MTOR |
P42345 |
MTOR_HUMAN |
Serine/threonine-protein kinase mTOR |
BioGRID |
Pubmed:15843522 |
https://thebiogrid.org/108757/summary/homo-sapiens/mtor.html |
https://thebiogrid.org/interaction/282428 |
|
4 |
RPS6KB1 |
P23443 |
KS6B1_HUMAN |
Ribosomal protein S6 kinase beta-1 |
BioGRID |
Pubmed:15843522 |
https://thebiogrid.org/112112/summary/homo-sapiens/rps6kb1.html |
https://thebiogrid.org/interaction/282430 |
|
5 |
TUBA1B |
P68363 |
TBA1B_HUMAN |
Tubulin alpha-1B chain |
BioGRID |
Pubmed:23741361 |
https://thebiogrid.org/115651/summary/homo-sapiens/tuba1b.html |
https://thebiogrid.org/interaction/872857 |
|
6 |
YWHAQ |
P27348 |
1433T_HUMAN |
14-3-3 protein theta |
BioGRID |
Pubmed:10835362 |
https://thebiogrid.org/116168/summary/homo-sapiens/ywhaq.html |
https://thebiogrid.org/interaction/306078/tert-ywhaq.html |
|
7 |
YWHAZ |
P63104 |
1433Z_HUMAN |
14-3-3 protein zeta/delta |
BioGRID |
Pubmed:10835362 |
https://thebiogrid.org/113366/summary/homo-sapiens/ywhaz.html |
https://thebiogrid.org/interaction/306076 |
In this revision, we have provided the detailed description of our database CBS Probe PINGSTM for protein-protein interaction, with the citation of our prior paper (reference #22) in the method section (page 10). We expect that these could help enhance the validity and reliability of our assessment using CBS Probe PINGSTM for PPI network analysis.
Q2) A pervasive problem in the manuscript is the use of “marginally significant” vs. significant results. While the authors can briefly discuss trends, marginally significant use is non-standard and without statistical meaning. The conclusions that ARE borne out by the data presented are restricted to the following.
- Figures 1C, 1D, and 1E clearly show that tumors are associated with promoter mutations (interesting), telomere length increases, and TERT expression increases.
- Differences in HCC are significant for TERT expression and telomere length. This was verified by univariate analysis.
- Overall survival IS affected by promoter mutations.; Progression-free survival IS different with low TERT vs. high TERT, a result that is not unexpected.
Ans) We thank the reviewer for the critical comment. One of the important drawbacks in studies on TERT so far is the lack of integrative analysis of telomere, telomerase, and related parameters and TERT alteration in patients with HCC. In addition, the prior studies have been all limited to patients undergoing surgical resection and mostly analyzed only one of TERT factors. Through the comprehensive analysis of TERT-related factors, we herein tried to integrate and better understand the respective roles that telomere, telomerase, and TERT alterations play in HCC patients receiving non-surgical treatments as well as surgical resection. Besides the aforementioned three points raised by the reviewer, we would like to provide the following additional points from our analyses:
(3) There are negative correlations between the expression statuses of the shelterin complex proteins, TERT, and telomere length (Figure 1C and Supplementary data)
(4) With tumor stage progression, the TERT level is significantly upregulated, whereas telomere length significantly decreases (Figure 2C).
(5) Regarding pathological tumor differentiation, TERT expression levels are positively correlated with HCC histological grade (Figure 2D).
(6) The predictive role of TERT promoter mutation differs with the etiology of HCC: TERT promoter mutation is a significant predictor of poor survival in patients with non-HBV-related HCC but not in those with HBV-related HCC (Supplement data).
(7) TERT mutations act as a biomarker for early-stage HCC, while TERT expression does for later-stage HCC.
Based on these findings, we have concluded that TERT and telomere alterations play a crucial role in all-stage liver carcinogenesis including the development and progression of HCC. TERT-telomere abnormalities might serve as useful biomarkers for HCC, but the prognostic values may differ with tumor characteristics and treatment
Our study has strengths recruiting the surgical cohort as well as non-surgical cohort, and those with early to advanced HCC, which represent a good study population to explore the entire scope of TERT-telomere biology in hepatocarcinogenesis, as compared to the study limited to the surgical cohort with early-stage HCC. We agree with the reviewer’s concern regarding statistical meaning. However, we believe that although some of the results within our data did not reach statistical significance, those may not necessarily indicate the lack of significance at all. Rather, our study findings suggest that it is worth further investigation regarding TERT-telomere biology across the wide spectrum of HCC disease. The true associations would be apparent with future studies involving a larger sample size. With all due respect to the reviewer’s comments, we have removed and minimized the reporting of ‘a marginal significance’ throughout the text.
Q3) An indication of the actual range of the absolute telomere length is needed to draw any inference regarding genetic stability.
Ans) In our experiment, the absolute telomere length was measured using reference human genomic DNA sample and ScienCell's Absolute Human Telomere Length Quantification qPCR Assay Kit (AHTLQ, Catalog #8918), as previously reported [Nathan O'Callaghan et al. Biotechniques 2008;44(6):807-9]. The reference genomic DNA sample with known telomere length was used as a reference for the calculation of telomere length of target samples (absolute telomere length is provided as 758 ± 36 kb per diploid cell or 8.24 ± 0.39 kb per chromosome end). The kit also provides a telomere amplification primer and another one that recognizes and amplifies a 100 bp-long region on human chromosome 17. In our study, telomere lengths were assessed in 86 evaluable samples. The absolute telomere length was 39.8 ± 2.9 (interquartile range: 10.79-110.41), with 55.9 ± 4.1 for the tumor and 8.8 ± 0.6 for the non-tumor samples. The long and short telomere length groups were classified according to the median value, as described in the Method section. As per the reviewer’s comment, we have clarified the method to measure absolute telomere length and indicated the actual range of the absolute telomere length in our samples (page 4).
Q4) Trivial textual issue: Need to define TACE., OS in the text in addition to the figure legend.
Ans) We have defined TACE in page 2 and figure 4 and also defined OS in page 6.
Submission Date 01 February 2021
Date of this review 25 Feb 2021 18:40:58

Round 2
Reviewer 1 Report
Authors have addressed concerns presented by the reviewers.
Author Response
there was no comment from the Review Report (Reviewer 1)
Reviewer 2 Report
General comments:
The authors have made many of the changes proposed by this reviewer.
Furthermore, the new findings with regard to the expression of shelterin TRF1, TRF2 and POT1 mRNA expression levels are quite interesting and may be now the most biologically significant finding that should spur further investigations. These results however are underexplored (and perhaps underappreciated). As a consequence, obvious experimental extensions of these studies are not conducted, and the findings are not discussed in depth.
Experimentally,
- All six components of shelterin must be assayed.
- The mRNA studies should be complemented by Western analysis for TRF1, TRF2 and POT1.
In terms of presentation,
- The results are presented of equal importance (and almost as an aside) to the negative PPI predicted interactor results. The latter studies serve primarily as a good control for the shelterin data.
- There is little discussion of the significance of the shelterin data. The authors should explore literature on the effect of shelterin component overproduction (e.g., inflammation). Consider also that a change in stoichiometry may alter the stability of complexes. I appreciate that the authors are more interested in biomarkers than in mechanism, but even at this level further exploration is necessitated.
Other issues:
Textual
Line 7, sentence is non-grammatical.
Line 37. The authors need to define “interacting” in a more nuanced fashion (e.g., network interactors, physical interactors, telomerase recruitment proteins, etc.).
Line 45. What is the evidence that the TERT telomere network HAS a substantive effect on HCC? Don’t you mean that it MAY have an effect?
Line 64. Define telomerase as an RNP-reverse transcriptase that uses its RNA for template for the addition of simple sequence DNA.
Line 66. Change “such as” to “including” and add abbreviations TPP1 and RAP1.
Line 105. Please provide some background on the nature and function of these proteins. How do they functionally interact? Don’t just rely on the ontology program.
Line 151. Change “marginally interact” to “may interact”
Line 224-225, Line 279-281. The authors need to discuss the potential significance of the results obtained at this point in the Discussion. The statements given are unnecessarily vague.
Author Response
Comments and Suggestions for Authors
General comments:
The authors have made many of the changes proposed by this reviewer.
Furthermore, the new findings with regard to the expression of shelterin TRF1, TRF2 and POT1 mRNA expression levels are quite interesting and may be now the most biologically significant finding that should spur further investigations. These results however are underexplored (and perhaps underappreciated). As a consequence, obvious experimental extensions of these studies are not conducted, and the findings are not discussed in depth.
Experimentally,
- Q) All six components of shelterin must be assayed.
Ans) We thank the reviewer for this point. As per the reviewer’s comment, we have tested all six components of shelterin, including TRF1, TRF2, TIN2, POT1, TPP1, and RAP1 in our samples. As a result, all shelterin components except TPP1 were significantly overexpressed in the non-tumors compared to the tumors. Consistent with the previous findings, there were overall negative correlations between the expression statuses of the shelterin complex proteins and TERT and telomere length, indicating that the shelterin complex functions as a negative regulator of telomerase. The results are provided in Line 118-122 and Figure 1C.
Q2) The mRNA studies should be complemented by Western analysis for TRF1, TRF2 and POT1.
Ans) We appreciate the reviewer’s point on the Western blot assay for the shelterin complex. Unfortunately, we are unable to produce the results of western blot in this revision within the timeline of the 10 day-revision process. Instead, we would like to provide the results from the published studies on this point. El Idrissi, et al., evaluating only POT1 and TRF2, showed cause-specific patterns of POT1 and TRF2 western blot in 40 patients with HBV-, HCV-, or alcohol-related HCC [J Exp Clin Cancer Res 2013;32:64.]. In this study, the results of Western-blot analysis were generally consistent with those of the qRT-PCR. Another study by Kim et al. examining tissue immunohistochemistry showed that TPP1, TRF2, RAP1, and POT1 correlated with HCCs expressing stemness-related markers [J Hepatol 2013;59:746.]. However, this study examined the shelterin complex only by Immunohistochemistry, not by Western-blot assay. To the best of our knowledge, no study to date has shown the data on both the qRT-PCR and Western-blot analysis of all six shelterin components in HCC patients. And no study has specifically evaluated the potential utility of shelterin complex as a biomarker in a large cohort of HCC patients. As stated in the manuscript, the aim of this study was to investigate the clinico-pathological association and biomarker function of TERT promoter mutation, its expression, and telomere length in HCC patients undergoing hepatectomy as well as TACE. We agree with the reviewer that the regulatory mechanism of shelterin in TERT and telomere function is important in cancer. However, based on the specific aims of the current study, we believe that a detailed analysis involving the qRT-PCR and Western analyses about shelterin and telomere dysfunction is likely beyond the scope of the study. These issues will be the focus of our future studies. However, with all due respect to the reviewer, we have commented this point in the limitation section and other appropriate sections in revised Discussion (Line 295-296, 225-229, and 282-288).
In terms of presentation,
- The results are presented of equal importance (and almost as an aside) to the negative PPI predicted interactor results. The latter studies serve primarily as a good control for the shelterin data.
Response) The negative PPI-predicted interactor results are well matched with the TERT data in our patients, and as pointed out by the reviewer they serve as a good control for the shelterin complex data in our results. We acknowledge that apart from the significance of telomere and TERT alterations in HCC, the shelterin complex can also influence liver carcinogenesis through the regulation of telomerase and telomere length. This should be further studied due to limited information in settings of HCC. We deeply appreciate the reviewer’s insightful comment in this review.
- Q) There is little discussion of the significance of the shelterin data. The authors should explore literature on the effect of shelterin component overproduction (e.g., inflammation). Consider also that a change in stoichiometry may alter the stability of complexes. I appreciate that the authors are more interested in biomarkers than in mechanism, but even at this level further exploration is necessitated.
Ans) We thank the reviewer for the comments on the shelterin complex. Through a search for relevant studies on shelterin, it was reported that the shelterin complex plays a crucial role in telomere and TERT regulation and has potential diagnostic and therapeutic implications in various cancers [multiple studies]. In addition, the shelterin and telomere dysfunction was associated with inflammation, aging, and metabolic diseases. Shelterin is formed through dynamic interactions between its protein subunits. Recently, reconstitution or a change in stoichiometry in shelterin complex was reported to alter telomerase processivity [Lim CJ et al. Nat Commun 2017;8:1075.]. Unfortunately, studies on shelterin are very scarce in HCC. In this revised version, we have added discussions in relation to our shelterin data to the appropriate sections of revised Discussion according to the reviewer’s suggestions (Line 225-229, 282-288, and 295-296,).
Other issues:
Textual
- Q) Line 7, sentence is non-grammatical.
Ans) We think that there is a typo, Line 7 should be line 17. We thank the reviewer for bringing this to our attention. We have made corrections to the sentence. It reads ‘Telomerase reverse transcriptase (TERT) mutations are the most frequent genetic alterations in hepatocellular carcinoma (HCC).’ on line 17, page 1.
- Q) Line 37. The authors need to define “interacting” in a more nuanced fashion (e.g., network interactors, physical interactors, telomerase recruitment proteins, etc.).
Ans) We thank the reviewer for this point. As per the reviewer’ request, we have defined “interacting” in the sentence. It reads ‘Protein-protein interaction (PPI) analysis was performed to identify a set of genes that physically interact with TERT.’.
- Q) Line 45. What is the evidence that the TERT telomere network HAS a substantive effect on HCC? Don’t you mean that it MAY have an effect?
Ans) We agree with the reviewer’s concern regarding the statement on Line 45. Although we herein presented some data on the effect of the TERT-telomere network on HCC, it still remains incompletely studied, and requires further investigation to substantiate its effect on HCC. Thus, we have made changes to the wording of ‘has’ to ‘may ~ ’ according to the reviewer’s suggestions (Line 45, page 1).
- Q) Line 64. Define telomerase as an RNP-reverse transcriptase that uses its RNA for template for the addition of simple sequence DNA.
Ans) We have defined telomerase in Line 45. It reads ‘Telomeres are elongated by telomerase, a ribonucleoprotein-reverse transcriptase complex that uses its RNA for template for the addition of simple telomeric repeats.’ (Line 64-66, page 2)
- Q) Line 66. Change “such as” to “including” and add abbreviations TPP1 and RAP1.
Ans) We have changed “such as” to “including” and added abbreviations TPP1 and RAP1 (Line 67-70, page 2), according to the reviewer’s suggestions.
- Q) Line 105. Please provide some background on the nature and function of these proteins. How do they functionally interact? Don’t just rely on the ontology program.
Ans) We thank the reviewer for this helpful comment. We have explored the nature, function, and interaction of the proteins and provided the summary of functional interactions between the proteins in Supplementary Table S1 (Line 111-112, page 3). The summary is provided with published data with citations, not relied on the ontology program.
- Q) Line 151. Change “marginally interact” to “may interact”
Ans) We have made changes to the sentence. It reads ‘In the tumors, the presence of TERT promoter mutation may be associated with an increasing level of TERT expression and telomere length, ..’ (Line 155, page 5)
- Q) Line 224-225, Line 279-281. The authors need to discuss the potential significance of the results obtained at this point in the Discussion. The statements given are unnecessarily vague.
Ans) We thank the reviewer for the helpful comments. We have deleted or made some modifications to the ambiguous sentences. It reads ‘These findings ~ in hepatocarcinogenesis’ (Line 228-229, page 8) and ‘Telomere length ~ further investigation’ (Line 282-288) in the appropriate sections of Discussion. We hope the revision would help clarify the intended meaning of the statement.
